# Preterm birth and neonatal mortality in selected slums in and around Dhaka City of Bangladesh: A cohort study

**Abdur Razzaque**[1]*, **Anisur Rahman**[1], **Razib Chowdhury**[1], **A. H. M. Golam Mustafa**[1], **Shakera Naima**[1], **Farzana Begum**[1], **Sohana Shafique**[1], **Bidhan Krishna Sarker**[1], **Mohammad Zahirul Islam**[2], **Minjoon Kim**[3], **Margub Aref Jahangir**[4], **Ziaul Matin**[5], **Jannatul Ferdous**[4], **Maya Vandenent**[4], **Daniel D. Reidpath**[1]

1 International Centre for Diarrheal Disease Research, Bangladesh (icddr,b), Dhaka, Bangladesh,
2 Embassy of Sweden, Dhaka, Bangladesh, 3 Maternal Newborn Health, UNICEF New York, New York, New York, United States of America, 4 UNICEF Bangladesh, Dhaka, Bangladesh, 5 Health Section, UNICEF India, New Delhi, India

* razzaque@icddrb.org

**Data Availability Statement:** All data files are available from the Harvard Dataverse and can be available at https://doi.org/10.7910/DVN/SMOF8P.

## Abstract

### Background

Although under-five mortality has declined appreciably in Bangladesh over the last few decades, neonatal mortality still remains high. The objective of the study is to assess the level and determinants of preterm birth and the contribution of preterm birth to neonatal mortality.

### Methods

Data for this study came from selected slums in and around Dhaka city, where; since 2015, icddr,b has been maintaining the Health and Demographic Surveillance System (HDSS). The HDSS data were collected by female Field Workers by visiting each household every three months; however, during the visit, data on the Last Menstrual Period (LMP) were also collected by asking each eligible woman to ascertain the date of conception. Gestational age was estimated in complete weeks by subtracting LMP from the date of the pregnancy outcome. In this study, 6,989 livebirths were recorded by HDSS during 2016–2018, and these births were followed for neonatal survival; both bivariate and multivariate analyses were performed.

### Results

Out of total births, 21.7% were born preterm (before 37 weeks of gestation), and sub-categories were: 2.19% for very preterm (28 to 31 weeks), 3.81% for moderate preterm (32 to 33 weeks), and 15.71% for late preterm (34 to 36 weeks). The study revealed that preterm babies contributed to 39.6% of neonatal deaths; however, the probability of death was very high on the 1st day of birth (0.124 for very preterm, 0.048 for moderate preterm, 0.024 for late preterm, and 0.013 for term birth), and continued until the 3rd day. In the regression

**Funding:** The study is funded by United Nation's Children Fund, Bangladesh (Grant number: 01867). The funder played a critical role in preparation of the manuscript.

**Competing interests:** The authors have declared that no competing interests exist.

**Abbreviations:** BDHS, Bangladesh Demographic Health Survey; BDT, Bangladeshi Taka; CI, Confidence Interval; HDSS, Health and Demographic Surveillance System; LMP, Last Menstrual Period; SDG, Sustainable Development Goal; SPSS, Statistical Program for Social Science; UNICEF, United Nations International Children's Emergency Fund; WHO, World Health Organization.

analysis, compared to the term neonates, the odds of neonatal mortality were 8.66 (CI: 5.63, 13.32, p<0.01), 4.13 (CI: 2.69, 6.34, p<0.01) and 1.48 (CI: 1.05, 2.08, p<0.05) respectively for very, moderate, and late preterm birth categories. The population attributable fraction for neonatal mortality was 23%, and sub-categories were 14% for very preterm, 10% for moderate preterm, and 6% for late preterm.

## Conclusions

Although urban slums are in proximity to many health facilities, a substantial proportion of preterm births contribute to neonatal deaths. So, pregnant women should be targeted, to ensure timely care during pregnancy, delivery, and post-partum periods to improve the survival of new-borns in general and preterm birth in particular.

## Introduction

Globally, under-five mortality rates although are improving, neonatal mortality rates have shown much less progress. In Bangladesh, neonatal mortality accounts for 69% deaths of infants and 66% deaths of under-five children [1]. To improve the situation, the Sustainable Development Goal 3 (SDG-3) has proposed to reduce under-five deaths further, and each country should target to reduce the neonatal mortality rate to at least 12 per 1,000 livebirths by 2030 [2].

The survival chance of a baby depends on gestation age. In fact, preterm is defined as babies born before 37 completed weeks of gestation, while categories of preterm births are: a) very preterm (28 to 31 weeks); b) moderate preterm (32 to 33 weeks); and c) late preterm (34 to 36 weeks); those born at 37 or more weeks of gestation were classified as term birth. Globally, an estimated 15 million babies are born preterm each year (more than one in 10 babies); two-fifth of these births were in south Asia and sub-Saharan Africa [3, 4]. At the national level, preterm birth is about 5% in developed countries, while it is 25% in low-medium-income countries [5]; however, the preterm birth rate is increasing in some developed countries [6].

Among the 13.4 million preterm births, 27% of them were died at neonatal period globally which made this one of the most common cause of death [7]. The risk of death of premature babies is at a higher level during the neonatal period than those born at term, especially in developing countries, where about half of all mothers give birth at home without a skilled birth attendant [7]. Complications of preterm births are the leading direct cause of neonatal mortality, accounting for approximately 1 million of almost 4 million neonatal deaths every year, and act as a risk factor for many neonatal deaths due to other causes, particularly infections [8]. Preterm babies who survive are at a higher risk for short-term and long-term morbidities [9–11]. In Bangladesh, 17.2% of newborn deaths are due to complications of premature births [12]. Preterm neonates also contribute significant costs to health systems, and families of preterm newborns often experience considerable psychological and financial hardships [13, 14].

The decrease in neonatal mortality of preterm birth in developed countries is primarily due to better management than changes in the percentage of preterm births [15–17]; postnatal care for preterm babies in developed countries uses specialized equipment and trained doctors. Low-cost and effective services appropriate to home births in low-income countries are rare [18, 19]. However, some studies from high-mortality settings have demonstrated that interventions targeting home care services by a skilled community health worker for providing service

to pregnant women during the antenatal, delivery, and postnatal periods reduced neonatal mortality significantly [20–26].

According to the last slum census in 2014, a total of 13,938 slums were counted, covering 2,227,754 residents in all cities and other urban areas of Bangladesh (for details, see BBS 2015 [27]). Out of those, 33.6% were counted in Dhaka North (11.8%), Dhaka South (12.6%), and Gazipur city corporation (9.2%). The Bangladesh Urban Health Survey reported that neonatal mortality is higher in the slum areas (43.7 per 1000 livebirths) than in non-slum areas (20.1 per 1000 live births). Reports also indicate the inequality of neonatal deaths regarding the location and size of the slums as well as sex of the child [28, 29].

Studies of preterm birth and neonatal mortality were few, particularly among slum dwellers in developing countries, and it is due to the scarcity of data. The urban HDSS has been in operation in selected slums in and around Dhaka city to monitor and impact evaluation of the comprehensive primary health care services that are being provided by the Aalo clinics (model clinics). The study took advantage of using high quality HDSS data of the slum area, those at great risk of experience of preterm birth; the study slums cover about one-third of slum population of Bangladesh. The objective of the study is to assess the level and determinants of preterm birth and the contribution of preterm birth to neonatal mortality using data from selected slums in and around Dhaka city.

## Methodology

### Study design and settings

The study used data from the urban Health and Demographic Surveillance System (HDSS) which operates on some purposively selected *slums* in Dhaka (North & South) and Gazipur City Corporations, where icddr,b has been maintaining the HDSS since 2015 (for detail, see *Slum Health in Bangladesh* [29]). We extracted data of a birth cohort from urban HDSS and followed for <29 days since birth. During the follow-up visits, each married woman (15–49 years) was asked privately by the female Field Worker to know whether they had been menstruating or not; if not, then asked about their Last Menstrual Period (LMP) to ascertain the conception status. Once the conception was confirmed, the woman was followed for subsequent pregnancy outcomes (livebirth, stillbirth, induce miscarriage, and spontaneous miscarriage). For each livebirth, data were collected on the date of delivery, place of delivery, details on maternity care (antenatal, delivery, and postnatal practice). In this study, all the livebirths in the study area for the period 2016–2018 were included and followed for survival up to the neonatal period. Refusal is rare, and the community is always been supportive of icddr,b activities, as they get free treatment for diarrhoea from Dhaka hospital.

All the inhabitants of those selected slums were interviewed for the baseline census in 2015 covering 118,238 population (3.8 persons per household). After the baseline census data collection, all the households are followed in every 3 months of interval. These data cover registration of conception, pregnancy outcomes (livebirths, stillbirths, induced miscarriage, and spontaneous miscarriage), deaths, marriage/divorce, migrations (in-, out-, and internal movement), changes in household composition/headship, safe motherhood practices. In these slums, most households had one bedroom (82%) with a small dwelling area (119 sq ft). Construction materials are usually brick/cement floors (88%), tin walls (70%), and tin roofs (94%). About 95% of households used pipe water for drinking, while only one-third had sanitary latrine flush to sewerage/septic tank. Sharing facilities are common in the slums, with 92% shared water sources, 90% latrines, and 60% cooking places; the use of electricity as a source of light is universal. Most households have electric fans (96%) and mobile phones (85%); only

60% had television and *khat*. The mean household expenditure was BDT 11,981 per month (US$ 145) and 72% of households did not have any savings [30].

## Data and measurements

Gestation age was estimated by subtracting the date of delivery from the date of LMP which was later converted into gestation weeks. Those born between 28 and 36 weeks of gestation were termed as preterm birth, and sub-categories were very preterm (28 to 31 weeks), moderate preterm (32 to 33 weeks), and late preterm (34 to 36 weeks); those born at 37 or more weeks of gestation were classified as term birth. Deliveries below 28 weeks of gestation age were considered either miscarriages or stillbirth, and were not included in this analysis. The dependent variable used in this study is neonatal mortality–the probability of dying after birth and before 29 days of life (died or alive).

## Data analysis

First, neonatal mortality was calculated for each gestation weeks and for gestation age categories. The relative risk of neonatal death and a 95% confidence interval were calculated using gestational week, while '40 weeks' was used as a reference; while for categories of gestation age, term births (37 to 42 weeks) used as a reference category. The daily death rate up to the neonatal period was calculated with the number of deaths on a given day for the number of newborns surviving on that day.

The population attributable fraction (percent of deaths that could be prevented if the complications of prematurity could be eliminated) was calculated for neonatal mortality, and a 95% confidence interval was calculated by categories of preterm births [31]. The formula for population attributable fraction is-

$$PAF = P_i \left( 1 - \frac{1}{RR_i} \right)$$

Pi is number of deaths per births and RR is the risk ration of ith group.

We performed two both unadjusted (Model-I) and adjusted (Model-II) logistic regression analyses to determine the effect of the gestational age on neonatal mortality. Women's age at birth, education, working status, sex of the newborn, number of antenatal visits, and delivery related factors- mode, place and attendants were adjusted in logistic regression analyses (Model-II). Multicollinearity has been tested using the multiple linear regression methods incorporating the variance inflation factor (VIF) and acceptable range (VIF<5.0) included in the final logistic model. The whole analyses were conducted in STATA windows 15.0 version (Stata.corp, TX) and paper was reported following the STROBE guidelines.

## Ethics approval and consent to participate

All the methods of this study were carried out in accordance with the Declaration of Helsinki. icddr,b's Institutional Review Board (IRB) approved the protocol of the study (PR# 15045). A written informed consent was obtained from all subjects and/or their legal guardian(s) prior to the data collection. The authors had access to information that could identify the individual participants during and after data collection.

## Results

During the period (2016–2018), there were 6,989 livebirths were registered in the HDSS area, and all of them were selected, and of them, 265 died during the neonatal period (Fig 1). The

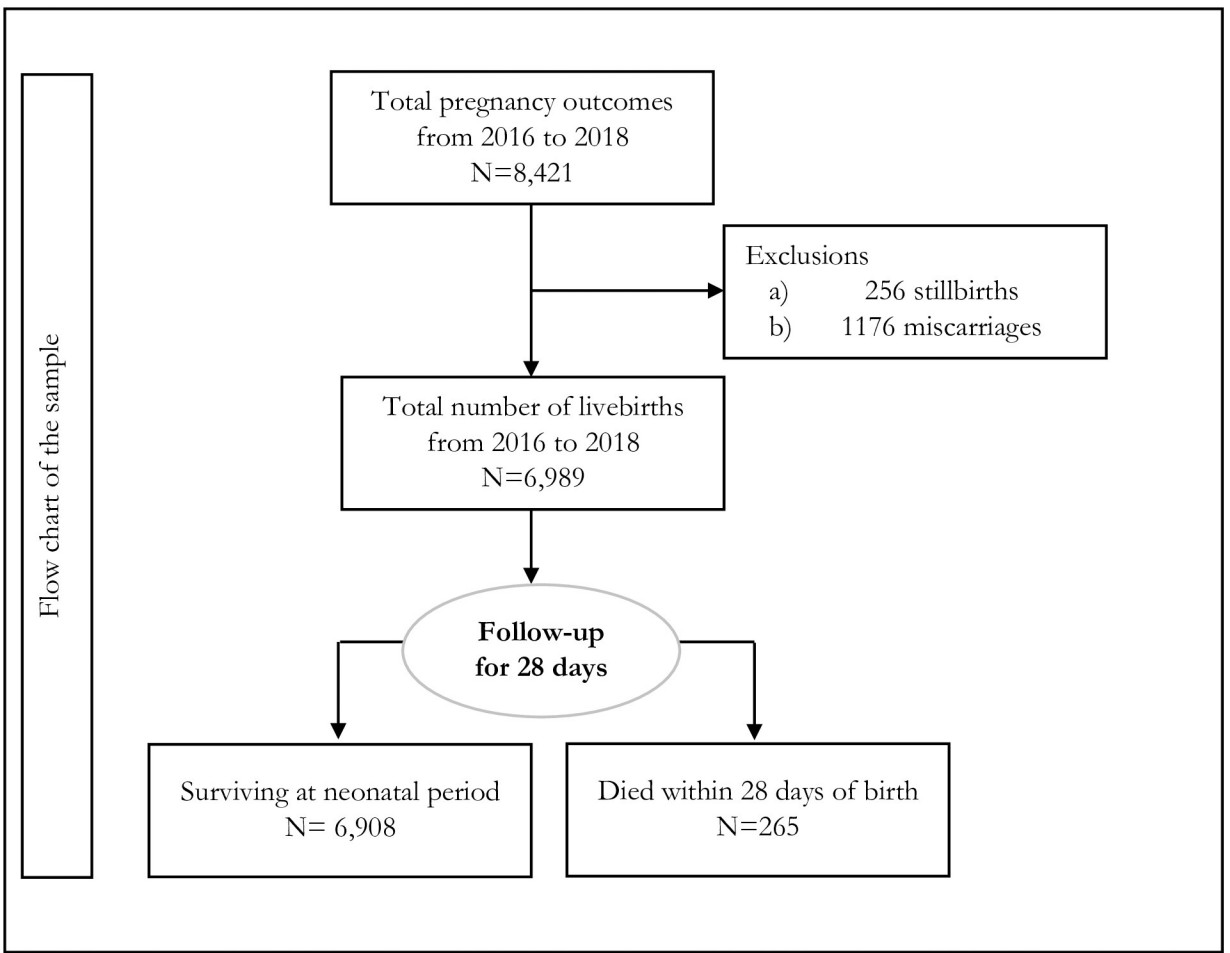

**Fig 1. Flowchart of the sample.**

average follow-up period was 27.03 days. During the follow-up period, none of the observations were missing as the culture of the postpartum period in the slum contexts. Women who had live births were usually not moved from their households.

Among them, most of the mothers gave birth at 18–24 years of age (54.6%) followed by the 25 or more years (36.7%), and only 8.7% of women gave birth at their adolescence (Table 1). Most of the mothers completed 5 or more years of schooling (50.3%), and 28% of the mothers had no formal education. During the study, nearly one-fourth of the women had worked outside (26.3%). Most of the mothers gave birth vaginally (72.3%) and with unskilled birth attendants (63.7%), though hospital delivery is higher than home delivery (51.4% vs. 48.6%). Most of the mothers received 1–3 antenatal care (ANC) visits (47.2%), followed by 4 or more ANC visits (36.8%), and 16% of the women had not received any ANC visits. Most of the births are term births (78.3%), and the rest are preterm births in different categories.

Both the number of births (0.20% to 1.32%) and the number of deaths (1.51% to 4.91%) were very low between 28 and 32 weeks of gestation (Table 2). From gestation age 34 weeks, the number of births started to increase gradually and reached from 255 (3.22%) to 671 (9.60%) at gestation age 37 and to 1513 (21.65%) at gestation age 39. Then the number of births declined gradually to 385 (5.51%) at gestation age 42. On the other hand, the number of deaths

**Table 1. Percent distribution of total sample.**

| Sociodemographic characteristics | Total | Survive | Death |
|---|---|---|---|
| | (N = 6989) | (N = 6724) | (N = 265) |
| **Mother's age (years)** | | | |
| <18 | 608 (8.7%) | 576 (94.7%) | 32 (5.3%) |
| 18–24 | 3816 (54.6%) | 3666 (96.1%) | 150 (3.9%) |
| 25 or more | 2565 (36.7%) | 2482 (96.8%) | 83 (3.2%) |
| **Years of schooling of mother** | | | |
| None | 1962 (28.1%) | 1877 (95.7%) | 85 (4.3%) |
| 1–4 | 1510 (21.6%) | 1446 (95.8%) | 64 (4.2%) |
| 5 or more | 3517 (50.3%) | 3401 (96.7%) | 116 (3.3%) |
| **Mother's occupation** | | | |
| Not working | 5148 (73.7%) | 4950 (96.1%) | 198 (3.9%) |
| Working | 1841 (26.3%) | 1774 (96.4%) | 67 (3.6%) |
| **Sex of Child** | | | |
| Male | 3580 (51.2%) | 3427 (95.7%) | 153 (4.3%) |
| Female | 3409 (48.8%) | 3297 (96.7%) | 112 (3.3%) |
| **Mode of delivery** | | | |
| Normal | 5053 (72.3%) | 4821 (95.4%) | 232 (4.6%) |
| Operation | 1936 (27.7%) | 19903 (98.3%) | 33 (1.7%) |
| **Birth attendant** | | | |
| Skilled | 2536 (36.3%) | 2471 (97.4%) | 65 (2.6%) |
| Unskilled | 4453 (63.7%) | 4253 (95.5%) | 200 (4.5%) |
| **Place of delivery** | | | |
| Home | 3399 (48.6%) | 3254 (95.7%) | 145 (4.3%) |
| Hospital | 3590 (51.4%) | 3470 (96.7%) | 120 (3.3%) |
| **ANC visits** | | | |
| No visits | 1117 (16.0%) | 1070 (95.8%) | 47 (4.2%) |
| 1–3 visits | 3301 (47.2%) | 3151 (95.5%) | 150 (4.5%) |
| 4 or more visits | 2571 (36.8%) | 2503 (97.4%) | 68 (2.6%) |
| **Preterm birth** | | | |
| Very preterm | 151 (2.2%) | 119 (78.8%) | 32 (21.2%) |
| Moderate preterm | 268 (3.8%) | 240 (89.6%) | 28 (10.4%) |
| Late preterm | 1097 (15.7%) | 1052 (95.9%) | 45 (4.1%) |
| Term birth | 5473 (78.3%) | 5313 (97.1%) | 160 (2.9%) |

was 12 (4.53%) at gestation age 34 and increased to 20 (7.55%) at gestation age 37, and to 39 (14.72%) at gestation age 39, then declined to 12 (4.53%) at gestation age 42 with a peak of 45 (16.98%) at gestation age 40. The relative risks of neonatal mortality were very high at gestation age 31 (4.02 times); however, there was an exceptionally high risk at gestation age 28 (9.92 times), followed by gestation age 30 (7.21 times). For gestation age categories 36 to 39 weeks and for categories 41 to 42 weeks, relative risks were lower than 1.00, which means that mortalities were lower at these ages than those recorded for gestation age 40 weeks.

Table 3 shows the number of births, number of deaths, neonatal death rates, relative risks, and population attributable fractions by gestation age categories. Both the numbers of births (2.19% and 3.81%) and deaths (12.08% and 10.57%) were low for gestation age categories of 28–31 and 32–33 weeks, but the number of births (15.71%) and the number of deaths (16.98%) increased from gestation age category 34–36 weeks, while both the numbers of births

**Table 2. Distribution of births, deaths, neonatal mortality rate and relative risk by gestational age.**

| Gestational age (weeks) | Number of birth (percent) | Number of death (percent) | NMR (per 1,000 livebirths) | RR (95% CI) |
|---|---|---|---|---|
| 28 | 14(0.20) | 5(1.88) | 357.1 | 9.92(1.75–21.64) |
| 29 | 19(0.27) | 4(1.51) | 210.5 | 5.84(2.33–14.61) |
| 30 | 50(0.72) | 13(4.91) | 260.0 | 7.21(4.16–12.49) |
| 31 | 69(0.99) | 10(3.77) | 144.9 | 4.02(2.11–7.63) |
| 32 | 92(1.32) | 8(3.02) | 87.0 | 2.41(1.17–4.96) |
| 33 | 174(2.49) | 20(7.55) | 114.9 | 3.19(1.93–5.27) |
| 34 | 225(3.22) | 12(4.53) | 53.3 | 1.48(0.79–2.75) |
| 35 | 392(5.61) | 20(7.55) | 51.0 | 1.41(0.84–2.36) |
| 36 | 481(6.88) | 13(4.91) | 27.0 | 0.75(0.40–1.37) |
| 37 | 671(9.60) | 20(7.55) | 29.8 | 0.82(0.49–1.38) |
| 38 | 890(12.73) | 21(7.92) | 23.6 | 0.65(0.39–1.09) |
| 39 | 1513(21.65) | 39(14.72) | 25.8 | 0.71(0.46–1.09) |
| 40[a] | 1249(17.86) | 45(16.98) | 36.0 | 1.00 |
| 41 | 765(10.95) | 23(8.68) | 30.1 | 0.83(0.50–1.36) |
| 42 | 385(5.51) | 12(4.53) | 31.2 | 0.86(0.46–1.61) |
| All | 6989 | 265 | 37.9 | - |

Note: CI = Confidence interval; NMR = Neonatal Mortality Rate; RR = Relative Risk;

[a] Reference category is 40 weeks.

(78.29%) and deaths (60.38%) reached to peak at gestation age category 37–42 weeks. The relative risks showed a similar pattern, higher at lower gestation age categories and lowest at gestation age categories 37–42 weeks. In fact, the mortality risk of very preterm births was 7.15 times, 3.60 times for moderate preterm births, and 1.40 times for late preterm births compared to term births; for overall preterm births (less than 37 weeks), the mortality risk was 2.37 times. The population attributable fraction for neonatal mortality was 14% for very preterm birth, 10% for moderate preterm birth, and reduced to 6% for late preterm birth. For overall preterm birth, the population attributable fraction was 23%.

The probability of death for each category of preterm birth as well as those of term births was very high on the day of birth (Fig 2). For very preterm birth, the probability of death on the day of birth was 0.124, 0.048 for moderate preterm, and 0.024 for late preterm; the probability of death was 0.013 for term birth. However, by the 3rd day of life, the probability of death had reduced by 70%–95% for these categories of birth. During 7–28 days of life, the probability

**Table 3. Number of births, number of deaths, neonatal death rates, relative risks, and population attributable fractions by gestation age categories.**

| Gestation age categories[+] | No. of birth (percent) | No. of death (percent) | NMR (per 1000 livebirths) | RR (95% CI) | PAF (95% CI) |
|---|---|---|---|---|---|
| Very preterm | 153(2.19) | 32(12.08) | 209.15 | 7.15(5.07–10.08) | 0.14(0.15–0.28) |
| Moderate preterm | 266(3.81) | 28(10.57) | 105.25 | 3.60(2.45–5.27) | 0.10(0.07–0.14) |
| Late preterm | 1098(15.71) | 45(16.98) | 40.98 | 1.40(1.01–1.93) | 0.06(0.03–0.05) |
| Preterm | 1517(21.71) | 105(39.63) | 69.21 | 2.37(1.87–3.01) | 0.23(0.18–0.43) |
| Term | 5472(78.29) | 160(60.38) | 29.23 | 1.00 | - |
| All | 6989 | 265 | 37.92 | - | - |

Note: CI = Confidence interval; NMR = Neonatal Mortality Rate; RR = Relative risk; PAF = Population Attributable Fraction; [+]Preterm birth (28 and 36 weeks): Very preterm (28 to 31 weeks), moderate preterm (32 to 33 weeks), and late preterm (34 to 36 weeks); Term birth (37 or more weeks).

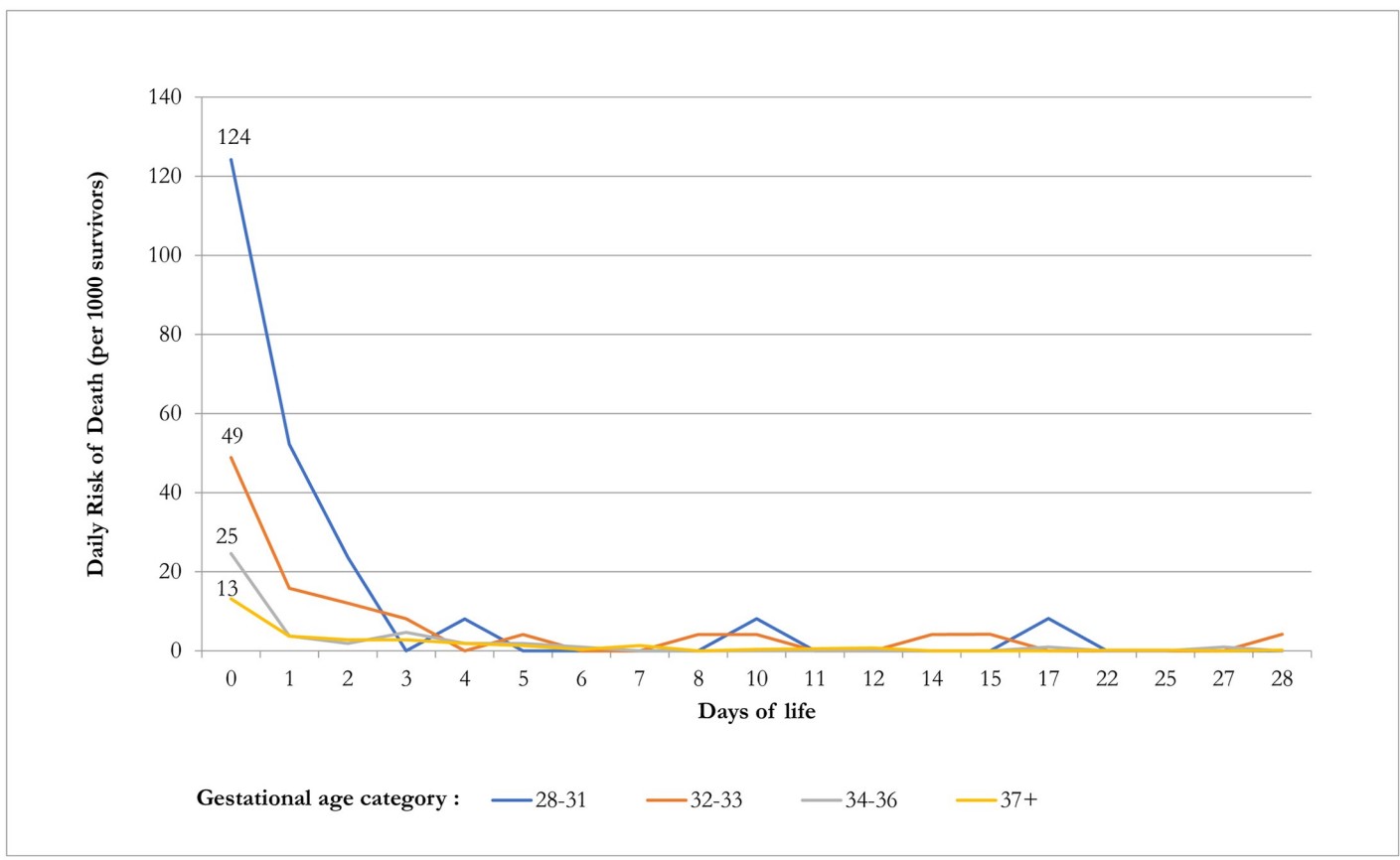

**Fig 2. Daily risk of death during the first month of birth by gestational age category.**

of death for term birth was 0.002, 0.016 for very preterm birth, 0.020 for moderate preterm birth, and 0.001 for late preterm birth; however, the number of cases for very preterm and late preterm birth were very few.

Table 4 shows regression estimates of neonatal mortality by preterm birth categories. Model-I included only preterm birth categories, while Model-II included preterm birth categories along with the selected socio-demographic variables as independent variables. For neonatal mortality, Model-I shows that the odds of death was 8.90 (CI: 5.84, 13.57; p<0.01) for very preterm birth, 3.98 (CI: 2.60, 6.07; p<0.01) for moderate preterm birth, and 1.42 (CI: 1.01, 1.99; p<0.05) for late preterm birth compared to term birth. Although the model II improved, the preterm birth was predominant over the model. The odds of death changed slightly, 8.66 (CI: 5.63, 13.32; p<0.01), 4.13 (CI: 2.69, 6.34; p<0.01) and 1.48 (CI: 1.05, 2.08; p<0.05) respectively for very, moderate and late preterm births.

Among the independent variables considered in model II, only sex of child (odds of death was 0.77 for female (CI: 0.59, 0.99; p<0.10)), age of mother (odds of death was 1.57 for aged <18 (CI: 1.01, 2.44; p<0.10)) and mode of delivery (odds of death was 2.80 for vaginal delivery (CI: 1.69, 4.64; p<0.01)) were found significant in this model.

## Discussion

This study analysed the prospective cohort data for estimating the effects of preterm birth on neonatal mortality in the urban poor (slum) area of Bangladesh. We used two models, where

**Table 4. Logistic regression estimates of neonatal mortality (N = 6908).**

| Variables | Model-I: Odd ratios, 95% CI | Model-II: Odd ratios, 95% CI |
|---|---|---|
| **Preterm birth** | | |
| Very preterm birth (rc = Term birth) | 8.90***(5.84, 13.57) | 8.66***(5.63, 13.32) |
| Moderate preterm birth (rc = Term birth) | 3.98***(2.60, 6.07) | 4.13***(2.69, 6.34) |
| Late preterm birth (rc = Term birth) | 1.42**(1.01, 1.99) | 1.48**(1.05, 2.08) |
| **Age of mother at birth (years)** | | |
| <18 (rc = 25 or more) | | 1.57*(1.01, 2.44) |
| 18–24 (rc = 25 or more) | | 1.25(0.94, 1.66) |
| **Sex of child** | | |
| Female (rc = Male) | | 0.77*(0.59, 0.99) |
| **Years of schooling of mother** | | |
| 1–4 (rc = None) | | 0.98(0.70, 1.39) |
| 5 or more (rc = None) | | 0.79(0.59, 1.06) |
| **Mother's working status** | | |
| Not working (rc = Working) | | 1.03(0.77, 1.38) |
| **Mode of delivery** | | |
| Vaginal (rc = Caesarean) | | 2.80***(1.69, 4.64) |
| **Place of delivery** | | |
| Home (rc = Hospital) | | 0.79(0.59, 1.07) |
| **Birth attendant** | | |
| Skilled (rc = Non-skilled) | | 1.03(0.68, 1.54) |
| **No. of antenatal visits** | | |
| 1–3 (rc = None) | | 1.16(0.82, 1.65) |
| 4 or more (rc = None) | | 0.76(0.51, 1.14) |
| **Constant** | 0.000 | 0.000 |
| -2 Loglikelihood (df) | 2149.53(3) | 2092.74 (14) |
| LR chi2 (p-value) | (0.000) | (0.000) |
| Pseudo $R^2$ | 0.051 | 0.080 |

Note: rc = Reference category; df = degrees of freedom; Model-I: preterm birth; Model-II: preterm birth along with selected socio-demographic and reproductive healthcare related factors; Very preterm = <32 weeks, moderate preterm = 32–33 weeks, late preterm = 34–36 weeks, 37 or more = term birth.

*indicates p<0.10,

** indicates p<0.05 and

*** indicates p<0.01

model-I showed the unadjusted effect of preterm birth on neonatal mortality and the model-II reflected it after adjusting other sociodemographic and reproductive factors. We found that every one out of five livebirths were preterm, which contributed to 39.% of neonatal deaths. The neonatal mortality rate was high (37.9 per 1000 livebirths) in the slum, and it varied considerably by different gestational ages. The probability of death for each category of preterm birth (very = 0.124, moderate = 0.048, and late = 0.024) and those of term birth (0.013) were very high on the day of birth; however, by the 3rd day of life, the probability of death reduced substantially. Compared to the term neonates, the odds of neonatal mortality were much higher for the lower gestational age, and the population attributable fraction reduced, respectively.

Estimation of gestational age based on reported LMP is usually criticised for inaccurate recall. In our study, the LMP data were collected by female Field Workers who visited each

household every three months to collect HDSS routine data, while each eligible woman was asked about their menstruation status secretly. Once conception was confirmed, LMP was recorded and the woman was followed prospectively for pregnancy outcome. Rahman et al. (2019) [32] estimated the gestational age based on LMP of ultrasound-based and LMP collected by female Field Workers of Matlab HDSS, and found that both estimates were close (Cronbach's alpha = 0.89). As conception data for urban HDSS is collected following a similar procedure as that for Matlab HDSS, so the urban HDSS data is believed to be of the same quality.

Our study found that 21.7% of births were preterm, which was close to that recorded (19.4%) by Baqui et al. (2013) [33], and the one recorded (22%) by Mamun et al. (2006) [34] from two different rural areas of Bangladesh. On the other hand, Rahman et al. (2019) [32] recorded a slightly higher level of preterm birth (29%) in Matlab HDSS data than our study for an earlier period (1990–1994), but for the recent period (2010–2014), the level of preterm birth has declined to almost half (11%).

Although a large number of studies have examined socio-demographic differentials of infant and under-five mortality in Bangladesh [35–37], none of these studies have specifically examined mortality differentials among slum dwellers. Recently, Razzaque et al. (2022) examined under-five mortality using the same data (HDSS) and found that it varied by the mother's working status, sex of the child, size of the baby, delivery type, presence of complication(s) during delivery, and birth interval [38]. Our current study found almost similar results when preterm birth was included in the model. Our study summarized that the preterm birth and being vaginally delivered are the strongest predictors and those had a higher chance of neonatal mortality. We also found a lesser but significant impact of maternal age less than 18 years and male sex of the newborn on increasing the risk of neonatal mortality. Therefore, raising awareness against adolescent pregnancy and the complications of preterm birth can reduce the risk of neonatal deaths in preterm births. Moreover, our study found caesarean delivery to be a life-saving intervention for preterm births. These findings reinforce the importance of awareness and access to reproductive and maternal health care to address infant mortality in poor urban settlements. Kannaujiya and others found that having a female child and caesarean delivery were less likely to die in the early neonatal period in India [39]. However, our adjusted factors had less impact on the overall model of explaining neonatal mortality than preterm births.

The neonatal mortality rate found in our study was about 80% higher (37.9 vs. 20.7 per 1000 livebirths) than that recorded in the Matlab HDSS government area [40]. In our study area, non-use of antenatal care was higher than those in the Matlab HDSS government area (12.0% vs. 6.3%), and a higher level of home delivery (49.0% vs. 37.6%) with a large proportion of pregnancy was attended by unskilled birth attendants (24.5% vs. 20.0%) [40, 41]. This has been occurring in the slum, although the study area is in proximity to many types of health facilities (public, private, and NGOs). In this regard, the World Health Organization (WHO) and the United Nations Children's Fund (UNICEF) jointly proposed that promotion of universal access to antenatal care, skilled birth attendance, and early postnatal care would contribute to a sustained reduction in maternal and neonatal mortality [42].

Our study found that one in five births are preterm (21.7%), while two in five preterm births died (39.6%) during the neonatal period. The neonatal mortality rate was 209.1 (per 1,000 livebirths) for very preterm birth, 105.2 for moderate preterm birth, and 41.0 for late preterm birth. The study also found that the population attributable fraction for neonatal mortality was 23%, which states that if excess deaths of preterm births could be prevented, there would be an almost one-quarter reduction of mortality at the population-level. The population attributable fraction found in our study was slightly lower (23% vs. 33%) than that recorded in another study from a rural area of Bangladesh [33]. The higher attributable fraction

documented by Baqui et al. (2008) [22] could be due to that their study area was under the intervention programme to reduce neonatal mortality, where home-based delivery was much higher than our study (91% vs. 48%).

The neonatal mortality rate by gestational age gives us insights into the mortality burden for preterm and term births. For very preterm birth, the neonatal mortality was high, but the proportion of death was low, while for moderate and late preterm birth, the neonatal mortalities were low, but the proportion of birth was high. Studies have reported that to improve the survival of preterm neonates, the need for very preterm births is quite different from that of moderate and late preterm births [7, 21, 33]. In this connection, Baqui et al. (2008) [22] reported that community-based strategies to manage newborn complications should focus on strengthening referral systems and reducing barriers to care-seeking to reach facilities quickly; however, these authors also mentioned that very little can be done for very preterm birth by community health workers in a rural community setting [7, 33]. In the HDSS, neonatal mortality was very high, although slums are in close proximity to many health facilities. So, intervention programme could be slightly different than those tested in rural settings. In the slum population, future intervention studies can be tested by targeting pregnant women along with strong referral linkages to facilities to ensure care during pregnancy, delivery, and post-partum periods; this might improve the survival of the new-borns in general and preterm birth in particular.

The strength of the study was that the data came from the HDSS area, which maintains high-quality data by visiting prospectively each household every three months and interviewing the respective women (for conception, delivery, and information of newborn). So, the date of occurrence of the event is believed to be fairly accurate. Besides, our study has made an important contribution by proving community-based preterm birth data from slum dwellers. The major limitation is the data of conception based on LMP, which is criticised for inaccurate recall; however, our record conception may not be a big issue as sufficient probing is done during the data collection and during the follow-up visit, it is rechecked. Additionally, the adjusted variables have less impact on the model, which limits the its ability explain neonatal mortality.

## Conclusions

Although urban slums are in proximity to many health facilities, neonatal mortality is still high in these slums, while a substantial proportion of preterm births contributed to neonatal deaths. So, pregnant women, especially adolescent mothers and vaginally delivered in the slum, should be targeted, along with strong referral linkage to facilities, to ensure care during pregnancy, delivery, and post-partum periods to improve the survival of new-borns in general and preterm birth in particular, in achieving the Sustainable Development Goal 3 for Bangladesh.

## Supporting information

**S1 Checklist. STROBE statement—Checklist of items that should be included in reports of observational studies.**
(DOCX)

## Author Contributions

**Conceptualization:** Abdur Razzaque, Anisur Rahman.

**Data curation:** Abdur Razzaque, Anisur Rahman, Razib Chowdhury, A. H. M. Golam Mustafa.

**Formal analysis:** Anisur Rahman, Razib Chowdhury, A. H. M. Golam Mustafa, Shakera Naima.

**Funding acquisition:** Abdur Razzaque.

**Investigation:** Abdur Razzaque.

**Methodology:** Abdur Razzaque, Anisur Rahman.

**Software:** A. H. M. Golam Mustafa.

**Supervision:** Abdur Razzaque.

**Writing – original draft:** Abdur Razzaque, Anisur Rahman.

**Writing – review & editing:** Razib Chowdhury, A. H. M. Golam Mustafa, Farzana Begum, Sohana Shafique, Bidhan Krishna Sarker, Mohammad Zahirul Islam, Minjoon Kim, Margub Aref Jahangir, Ziaul Matin, Jannatul Ferdous, Maya Vandenent, Daniel D. Reidpath.

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
