## [Decision Letter · Decision Letter 0]

6 Jun 2023

PONE-D-23-08400Preterm Birth and Neonatal Mortality in Selected Slums in and around Dhaka City of Bangladesh: A Cohort StudyPLOS ONE

Dear Dr. Razzaque,

Thank you for submitting your manuscript to PLOS ONE. After careful consideration, we feel that it has merit but does not fully meet PLOS ONE’s publication criteria as it currently stands. Therefore, we invite you to submit a revised version of the manuscript that addresses the points raised during the review process.

We look forward to receiving your revised manuscript.

Kind regards,

Rornald Muhumuza Kananura, PhD

Academic Editor

PLOS ONE

Journal Requirements:

3. Please expand the acronym “UNICEF” (as indicated in your financial disclosure) so that it states the name of your funders in full.

"icddr,b acknowledges with gratitude the commitment from UNICEF/Sida for continuation of these research efforts. icddr,b is also grateful to the Government of Bangladesh for its long-term financial support and also to international core donors: Canada, Sweden and the United Kingdom."

"The study is funded by United Nation’s Children Fund, Bangladesh (Grant number: 01867). The funder played a critical role in preparation of the manuscript."

Reviewers' comments:

Reviewer's Responses to Questions

**Comments to the Author**

1. Is the manuscript technically sound, and do the data support the conclusions?

Reviewer #1: Yes

Reviewer #2: Yes

2. Has the statistical analysis been performed appropriately and rigorously? 

Reviewer #1: Yes

Reviewer #2: Yes

3. Have the authors made all data underlying the findings in their manuscript fully available?

Reviewer #1: No

Reviewer #2: Yes

4. Is the manuscript presented in an intelligible fashion and written in standard English?

Reviewer #1: Yes

Reviewer #2: Yes

5. Review Comments to the Author

Reviewer #1: The authors have presented an analysis of health and demographic surveillance data on nearly 7000 births between 2016 and 2018 in Bangladesh, focusing on better understanding the frequency of preterm delivery and its potential predictors. This is an important topic, and the dataset used in this research is unique in its focus on slum neighborhoods. My suggestions for improvement are as follows:

Macro:

The authors state in the discussion that “Our study found that adolescent women who gave birth to a baby boy by normal delivery had a higher risk of neonatal mortality than compared to preterm delivery”. This statement confused me.

I read Table 4 as: “The strongest predictors of neonatal mortality among the variables assessed were preterm delivery, maternal age less than 18, male sex of the baby, and vaginal delivery.”

The odds ratios associated with preterm delivery were much higher than any of the others, which suggests that preterm delivery is a far bigger risk factor for neonatal mortality than any of the other significant variables… which suggests to me that emphasizing adolescent births, or male gender, or vaginal delivery isn’t the whole story.

This is also in the context of a model with an R2 of 0.08, which is extremely low. So I am not sure how much emphasis ought to be placed on the definitive nature of these findings.

The more important aspect of these findings is the timing of deaths and the different risks across categories of prematurity - which is appropriately emphasized, but I think care is necessary in interpreting the findings from Table 4.

Preterm delivery is complicated – and I think it’s important that the authors are careful to communicate that this analysis attempts to quantify differences in the timing of neonatal mortality associated with premature delivery, and while risk factors that prove to be significant are part of the story, many of them are correlated with one another and/or are not as important as prematurity itself.

The discussion could benefit from a more nuanced discussion of the implications of these findings.

This statement in the discussion also confused me:

Raising awareness about late pregnancy and caesarean delivery reduces the risk of neonatal deaths in preterm births.

Micro:

I might suggest different terminology on a few of the variables.

E.g.

- Normal delivery ought to be referred to as vaginal delivery ; “operation” ought to be referred to as caesarean section delivery

- sex = male / female rather than boy/girl

Table 1 – move the ‘total’ column to the left so you read OVERALL, then survival and death columns

Table 1 – To me, it makes sense to have column percentages rather than row percentages if you are trying to illustrate the differences by those who died vs. those who survived. Since the number who died is so small, row percentages are always going to make it difficult to see the real differences across categories within the predictor variables among babies who died. But this may be an issue of personal preference.

Reviewer #2: General Comments:

The study assessed the level and determinants of preterm birth and the contribution of preterm birth to neonatal mortality. I would like to thank the authors for this interesting study and the analysis they conducted. This paper addresses an important topic. I do not have any major criticisms: However, I have some concerns or suggestions that the authors may want to clarify.

Specific Comments

1. Did the authors have some pregnancy outcomes in their data that they were not able to register the pregnancies to asses to assess their Last Menstrual Period (LMP) before delivery? If yes, how were they handled?

2. Were there missing values? If yes, how did you handle them?

3. Check and clarify the statement from lines 67-69 “Preterm birth complications are estimated to be responsible for 35% of the world’s 3·1 million annual neonatal deaths, and are now the second most common cause of death after the economic cost of preterm birth is high in terms of neonate.”

4. Lines 269-269: The statement “Our study has made an important contribution by proving community-based preterm birth data from the slum dwellers” appears to be wrongly placed. I’m not sure what the intent is. The authors may want to clarify this.

5. Lines 276-277: The statement “Our study indicates that the adolescent women who gave birth to a baby boy by normal delivery had higher risks of neonatal mortality compared to preterm birth” is unclear. The interpretation of the results should be done carefully. Children born to adolescent mothers had increased odds of neonatal death compared to children born to mothers aged 25 or more years. Boys had increased odds of dying as neonates compared to girls, and normal deliveries were associated with increased odds of neonatal death compared to caesarian deliveries. You may want to revise your statement to clarify your statement.

6. Lines 277-279: You may want to clarify or revise the statement “Therefore, raising awareness about late pregnancy and caesarean delivery reduces the risk of neonatal deaths in preterm births.”

6. PLOS authors have the option to publish the peer review history of their article (what does this mean?). If published, this will include your full peer review and any attached files.

Reviewer #1: No

Reviewer #2: No

---

## [Author Response · Author response to Decision Letter 0]

31 Aug 2023

Response to Reviewers Comments

Reviewer #1: 

The authors have presented an analysis of health and demographic surveillance data on nearly 7000 births between 2016 and 2018 in Bangladesh, focusing on better understanding the frequency of preterm delivery and its potential predictors. This is an important topic, and the dataset used in this research is unique in its focus on slum neighborhoods. My suggestions for improvement are as follows

1. The authors state in the discussion that “Our study found that adolescent women who gave birth to a baby boy by normal delivery had a higher risk of neonatal mortality than compared to preterm delivery”. This statement confused me.

I read Table 4 as: “The strongest predictors of neonatal mortality among the variables assessed were preterm delivery, maternal age less than 18, male sex of the baby, and vaginal delivery.”

Authors response: We have changed the sentence in the discussion section according to your suggestion. Please read line number 284-287 “Our study summarized that the preterm birth and being vaginally delivered are the strongest predictor and those had higher chance of neonatal mortality. We also found a lesser but significant impact of maternal age less than 18 years and male sex of the newborn on increasing the risk of neonatal mortality.”

2. The odds ratios associated with preterm delivery were much higher than any of the others, which suggests that preterm delivery is a far bigger risk factor for neonatal mortality than any of the other significant variables… which suggests to me that emphasizing adolescent births, or male gender, or vaginal delivery isn’t the whole story.

This is also in the context of a model with an R2 of 0.08, which is extremely low. So I am not sure how much emphasis ought to be placed on the definitive nature of these findings.

The more important aspect of these findings is the timing of deaths and the different risks across categories of prematurity - which is appropriately emphasized, but I think care is necessary in interpreting the findings from Table 4.

Authors response: Our main objective of this study is to assess the association of preterm birth on neonatal mortality. Results (specifically table 4 and its interpretation) focused on this. Other sociodemographic and relevant independent variables are adjusted in the regression model II. However, we acknowledged that compared to model I, the variance explained (R2) of model II is not higher and comparatively low to report the fitness of the model. We made a slight change in results, please read line number 234 to 241: “Although the model II improved, the preterm birth was predominant over the model. The odds of death changed slightly, 8.66 (CI: 5.63, 13.32; p<0.01), 4.13 (CI: 2.69, 6.34; p<0.01) and 1.48 (CI: 1.05, 2.08; p<0.05) respectively for very, moderate and late preterm births. 

Among the independent variables considered in model II, only sex of child (odds of death was 0.76 for female (CI: 0.59, 0.99; p<0.10)), age of mother (odds of death was 1.57 for aged <18 (CI: 1.01, 2.44; p<0.10)) and mode of delivery (odds of death was 2.80 for vaginal delivery (CI: 1.69, 4.64; p<0.01)) were found significant in this model.”

We have added this as a limitation at the end of the discussion section; please read line number 332 and 333 “Additionally, the adjusted variables have less impact on the model which limits the model to explain the neonatal mortality.”

3. Preterm delivery is complicated – and I think it’s important that the authors are careful to communicate that this analysis attempts to quantify differences in the timing of neonatal mortality associated with premature delivery, and while risk factors that prove to be significant are part of the story, many of them are correlated with one another and/or are not as important as prematurity itself.

The discussion could benefit from a more nuanced discussion of the implications of these findings.

Authors response: In the result and discussion section we have mentioned the less impact of the adjusted variables, please read line number 234-235 in result section “Although the model II improved, the preterm birth was predominant over the model.”

4. And line number 295-296 in discussion section “However, our adjusted factors had less impact on the overall model of explaining neonatal mortality than the preterm birth.”

This statement in the discussion also confused me:

Raising awareness about late pregnancy and caesarean delivery reduces the risk of neonatal deaths in preterm births.

Authors response: We have changed the statements according to your suggestions. Please read line number 289-291 “Therefore, raising awareness against adolescent pregnancy and complications of preterm birth can reduce the risk of neonatal deaths in preterm births. Moreover, our study found caesarean delivery as a life-saving intervention for preterm births.”

5. I might suggest different terminology on a few of the variables. E.g.

- Normal delivery ought to be referred to as vaginal delivery ; “operation” ought to be referred to as caesarean section delivery; - sex = male / female rather than boy/girl

Authors response: We have changed accordingly. Please see table 1 and 4 as well as texts in texts accordingly.

6. Table 1 – move the ‘total’ column to the left so you read OVERALL, then survival and death columns

Authors response: We moved the total column to the left according to your suggestion. Please see table 1.

7. Table 1 – To me, it makes sense to have column percentages rather than row percentages if you are trying to illustrate the differences by those who died vs. those who survived. Since the number who died is so small, row percentages are always going to make it difficult to see the real differences across categories within the predictor variables among babies who died. But this may be an issue of personal preference.

Authors response: We acknowledged your concern. But, we are generally focusing on to illustrate the differences of survival and died children by different sociodemographic and other independent variables. So, we prefer the row percentage instead of the column percentage. I hope this will not affect the quality of the article’s results.

Reviewer #2: 

General Comments:

The study assessed the level and determinants of preterm birth and the contribution of preterm birth to neonatal mortality. I would like to thank the authors for this interesting study and the analysis they conducted. This paper addresses an important topic. I do not have any major criticisms: However, I have some concerns or suggestions that the authors may want to clarify.

Authors response: Thank you for your supportive feedbacks and comments. We have revised the manuscript according to your suggestion and the changes are made in the track changes form as well as mentioned below in each comment.

Specific Comments

1. Did the authors have some pregnancy outcomes in their data that they were not able to register the pregnancies to asses to assess their Last Menstrual Period (LMP) before delivery? If yes, how were they handled?

Authors response: According to the Urban HDSS resident, individual who lived in the HDSS area are concerned respondents. Mothers who were living in the HDSS area at the time of conception through pregnancy outcome were included in this study. And our female Field Workers, Supervisors and Data management team continuously worked on those HDSS population including mothers. Therefore, there is no chance of un-registered any of the pregnancy. Besides, the fame of icddr,b and our community engagements are quite strong that people in HDSS area are rarely refuse to be un-registered. We mentioned in line number 115-116.

2. Were there missing values? If yes, how did you handle them?

Authors response: In this study data, there is no missing values. As previously mentioned, due to strong community engagement and welfare fame of icddr,b the respected HDSS inhabitants are hardly refuse to provide information. Which lead to a greater good for the study.

3. Check and clarify the statement from lines 67-69 “Preterm birth complications are estimated to be responsible for 35% of the world’s 3·1 million annual neonatal deaths, and are now the second most common cause of death after the economic cost of preterm birth is high in terms of neonate.”

Authors response: We have revised the statements. Please read line number 67-70 “Among the 13.4 million preterm births, 27% of them were died at neonatal period globally which made this one of the most common cause of death [7].”

4. Lines 269-269: The statement “Our study has made an important contribution by proving community-based preterm birth data from the slum dwellers” appears to be wrongly placed. I’m not sure what the intent is. The authors may want to clarify this.

Authors response: We have shifted the statements in the strengths paragraph at the end of the manuscript. See line number 335 and 336.

5. Lines 276-277: The statement “Our study indicates that the adolescent women who gave birth to a baby boy by normal delivery had higher risks of neonatal mortality compared to preterm birth” is unclear. The interpretation of the results should be done carefully. Children born to adolescent mothers had increased odds of neonatal death compared to children born to mothers aged 25 or more years. Boys had increased odds of dying as neonates compared to girls, and normal deliveries were associated with increased odds of neonatal death compared to caesarian deliveries. You may want to revise your statement to clarify your statement.

Authors response: Thank you for the feedbacks. We have changed the statements according to your suggestions. Please see line number 284-287 “Our study summarized that the preterm birth and being vaginally delivered are the strongest predictor and those had higher chance of neonatal mortality. We also found a lesser but significant impact of maternal age less than 18 years and male sex of the newborn on increasing the risk of neonatal mortality.”

6. Lines 277-279: You may want to clarify or revise the statement “Therefore, raising awareness about late pregnancy and caesarean delivery reduces the risk of neonatal deaths in preterm births.”

Authors response: We have changed the statements according to your suggestions. Please read line number 289-291 “Therefore, raising awareness against adolescent pregnancy and complications of preterm birth can reduce the risk of neonatal deaths in preterm births. Moreover, our study found caesarean delivery as a life-saving intervention for preterm births.”

---

## [Decision Letter · Decision Letter 1]

27 Dec 2023

Preterm Birth and Neonatal Mortality in Selected Slums in and around Dhaka City of Bangladesh: A Cohort Study

PONE-D-23-08400R1

Dear Dr. Razzaque,

We’re pleased to inform you that your manuscript has been judged scientifically suitable for publication and will be formally accepted for publication once it meets all outstanding technical requirements.

Kind regards,

Rornald Muhumuza Kananura, PhD

Academic Editor

PLOS ONE

Additional Editor Comments (optional):

Reviewers' comments:

Reviewer's Responses to Questions

**Comments to the Author**

1. If the authors have adequately addressed your comments raised in a previous round of review and you feel that this manuscript is now acceptable for publication, you may indicate that here to bypass the “Comments to the Author” section, enter your conflict of interest statement in the “Confidential to Editor” section, and submit your "Accept" recommendation.

Reviewer #1: All comments have been addressed

2. Is the manuscript technically sound, and do the data support the conclusions?

Reviewer #1: Yes

3. Has the statistical analysis been performed appropriately and rigorously? 

Reviewer #1: Yes

4. Have the authors made all data underlying the findings in their manuscript fully available?

Reviewer #1: Yes

5. Is the manuscript presented in an intelligible fashion and written in standard English?

Reviewer #1: Yes

6. Review Comments to the Author

Reviewer #1: I am satisfied with the revisions as made and submitted by the authors of this manuscript. Well done.

7. PLOS authors have the option to publish the peer review history of their article (what does this mean?). If published, this will include your full peer review and any attached files.

Reviewer #1: No

---

## [Editor Report · Acceptance letter]

8 Jan 2024

PONE-D-23-08400R1 

PLOS ONE

Dear Dr. Razzaque, 

I'm pleased to inform you that your manuscript has been deemed suitable for publication in PLOS ONE. Congratulations! Your manuscript is now being handed over to our production team.

Kind regards, 

on behalf of

Dr. Rornald Muhumuza Kananura 

Academic Editor

PLOS ONE